# REBORN: Reinforcement-Learned Boundary Segmentation with Iterative Training for Unsupervised ASR

**Liang-Hsuan Tseng**\*   **En-Pei Hu**\*   **Cheng-Han Chiang**   **Yuan Tseng**
**Hung-yi Lee**   **Lin-shan Lee**   **Shao-Hua Sun**
National Taiwan University

## Abstract

Unsupervised automatic speech recognition (ASR) aims to learn the mapping between the speech signal and its corresponding textual transcription without the supervision of paired speech-text data. A word/phoneme in the speech signal is represented by a segment of speech signal with variable length and unknown boundary, and this segmental structure makes learning the mapping between speech and text challenging, especially without paired data. In this paper, we propose REBORN, **Re**inforcement-Learned **Bo**undary Segmentation with Ite**r**ative Trai**n**ing for Unsupervised ASR. REBORN alternates between **(1)** training a segmentation model that predicts the boundaries of the segmental structures in speech signals and **(2)** training the phoneme prediction model, whose input is the speech feature segmented by the segmentation model, to predict a phoneme transcription. Since supervised data for training the segmentation model is not available, we use reinforcement learning to train the segmentation model to favor segmentations that yield phoneme sequence predictions with a lower perplexity. We conduct extensive experiments and find that under the same setting, REBORN outperforms all prior unsupervised ASR models on LibriSpeech, TIMIT, and five non-English languages in Multilingual LibriSpeech. We comprehensively analyze why the boundaries learned by REBORN improve the unsupervised ASR performance.

## 1   Introduction

Automatic speech recognition (ASR) systems convert speech signals into their transcription texts. Most state-of-the-art ASR systems are trained with dense supervision using a large amount of labeled speech-text paired data [10, 52]. However, more than 80% of languages in the world have limited access to speech-text paired data [51], and collecting such paired data requires intensive labor and costs, fundamentally limiting the applicability of supervised ASR systems to these languages. This leads to significant efforts devoted to developing unsupervised ASR (UASR) systems, which aim to learn the mapping between speech signals and textual transcriptions (words or phonemes) without any speech-text pairs [4, 11, 18, 30, 33].

UASR models learn to align the distribution of input speech signal and output text without paired data. Learning to match two distributions of sequences unsupervisedly has been extensively studied in unsupervised neural machine translation (NMT) [1], where the aim is to learn a neural network that can translate text in a source language to text in a target language without paired data [26, 27]. Prior works show that adversarial training [20] can be used to learn such mappings [26, 27], which employs a generator learning to translate, and a discriminator learning to distinguish generated text from the real text in the target language.

---

\*Equal contribution.   Correspondence to: Shao-Hua Sun `<shaohuas@ntu.edu.tw>`

38th Conference on Neural Information Processing Systems (NeurIPS 2024).

Can we adopt such an adversarial training scheme for UASR? In unsupervised NMT, text can be easily tokenized into sub-word tokens, so unsupervised NMT only needs to learn the mapping between the source and target language's token embeddings. However, in UASR, a phoneme or word is represented by a variable-length segment in the speech signal whose boundaries are unknown. Moreover, the length of a speech signal is much longer than the length of its textual transcription. The above characteristics of speech make learning the mapping between the segmental structures in speech and textual transcription challenging. Existing works in unsupervised ASR rely on handcrafted rules or separately learned modules to obtain the boundaries of the segmental structures [4, 33, 64]. Yet, such handcrafted boundaries are often sub-optimal, bottlenecking the performance of UASR.

This paper focuses on learning better segmental boundaries to improve the mapping between segmented speech and textual transcription. We propose **REBORN** (**Re**inforcement-Learned **Bo**undary Segmentation with Ite**r**ative Trai**n**ing), an unsupervised ASR framework with a segmentation model and a phoneme prediction model. The segmentation model determines segmental structure boundaries in speech signals, while the phoneme prediction model assigns a phoneme to each segmental structure. After properly initializing the phoneme prediction model, we use an iterative algorithm that alternates between two stages to train the segmentation model and refine the phoneme prediction model. The first stage trains the segmentation model. Since we do not have the ground truth segmental boundaries, we use reinforcement learning to train the segmentation model to favor segmentations that yield phoneme sequence predictions with a lower perplexity. We experiment with various learning objectives and implement them as reward functions for learning the segmentation model. In the second stage, based on the boundaries predicted by the segmentation model learned in the previous stage, we use an adversarial loss similar to generative adversarial networks (GANs) [20] to train the phoneme prediction model.

We conduct extensive experiments on LibriSpeech [45], TIMIT [19], and Multilingual LibriSpeech (MLS) [50] to compare the phoneme/phone error rate (PER) and word error rate (WER) with prior works. We show that our method outperforms all prior UASR methods on LibriSpeech, TIMIT, and five languages in MLS when using the same amount of training data. We perform thorough ablation studies to show that iterative training and the rewards we design are critical to the performance of REBORN. By analyzing the segmental structure obtained by our segmentation model, we find that the segmental structures are acoustic units smaller than phonemes, which helps the phoneme prediction model predict more accurate transcriptions. To facilitate further research and replication, our code and models are available at `https://github.com/andybi7676/reborn-uasr`.

## 2  Related Work

An ASR model takes speech signals as the input and predicts the textual transcriptions. Unsupervised ASR (UASR) aims to train an ASR model without access to paired speech-text data. Instead, the only available training data is unlabeled speech data and unlabeled text data, while the correspondence between speech and text is not available. In this paper, we follow prior UASR works [4, 33] to predict phoneme transcriptions from speech signals. To achieve this, a lexicon is required to transform the unlabeled text into phoneme sequences. Learning the mapping between speech signals and phoneme sequences can be formulated as a distribution matching problem, where we want to learn an ASR model whose output phoneme sequences match the distribution of real phoneme sequences.

Yeh et al. [64] address this distribution matching problem using an unsupervised loss function based on empirical output distribution matching [34], which guides the ASR model to produce phoneme sequences with statistical distributions close to real phoneme sequences. Wang et al. [60] further extend this approach by matching the $N$-skipgram and positional unigram distributions.

Liu et al. [33] propose to use the generative adversarial network (GAN) [20] for UASR. The GAN solves the distribution matching problem by training a generator whose output distribution resembles a target distribution, and a discriminator is trained to distinguish the outputs of the generator and the samples from the target distribution. When using GAN to train unsupervised ASR, the generator is the phoneme prediction model. The phoneme prediction model takes in speech features of the segmental structures in the speech signal, and outputs a phoneme transcription. Prior works rely on hand-crafted or separately trained unsupervised phoneme segmentation models to find the boundary of the segmental structures [11, 33, 61].

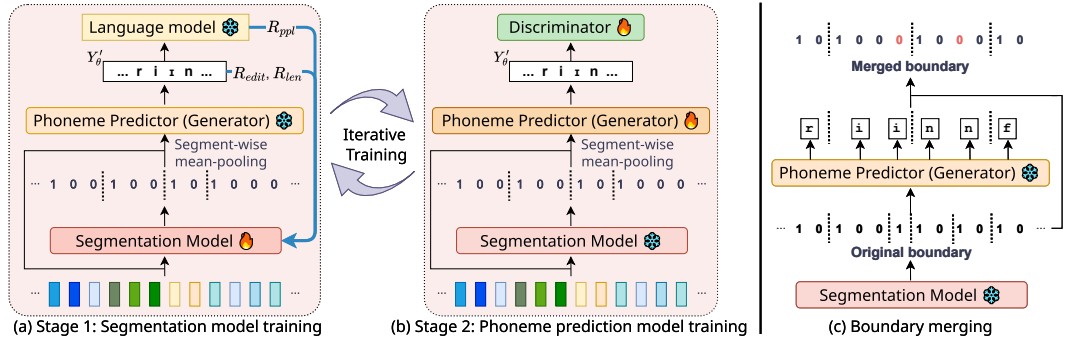

Figure 1: (a) and (b): REBORN iterates between using RL to train the segmentation model and using adversarial training to train the phoneme prediction model. (c): An illustration of the segmentation/boundary merging. 1 means the start of a segment while 0 is not. Given the original segmentation and the predicted phoneme sequence, we merge the segments that result in the same phoneme prediction into the same segment, yielding the merged boundary.

With the emergence of foundation models, wav2vec-U [4] shows that UASR performance can greatly benefit from using speech foundation models as the feature extractor. However, wav2vec-U still relies on the two-stage feature preprocessing step based on $k$-means to find the boundaries of the segmental structure in the speech signal. In our paper, we call the segmentation boundaries obtained by the feature preprocessing in wav2vec-U as "$k$-means-based segmentation". EURO [18] changes the decoding algorithm used to decode the phoneme prediction sequences and explores other speech foundation models for feature extraction, including HuBERT [23] and WavLM [12].

In these prior methods, the segmentation is either non-trainable or learned separately from the phoneme prediction model. REBORN differs from prior works by introducing a trainable segmentation model tailored for the phoneme prediction model, and the segmentation model and phoneme prediction model can be iteratively polished. wav2vec-U 2.0 [30] simplifies the feature preprocessing step in wav2vec-U and does not explicitly consider the segmentation structure in their model. RE-BORN has performance better than wav2vec-U 2.0 on almost all datasets, showing that learning the segmentation boundary for feature preprocessing is important for the performance of UASR.

Although recent UASR is originally to be a cross-modality distribution matching problem, it is also related to representation learning. For example, the speech features might be extracted from self-supervised speech foundation models; and the segmentation problem is also commonly investigated in tasks like acoustic unit discovery. We discuss these two related topics further in Appendix D.

## 3 Method: REBORN

The difficulty of mapping speech signals to their corresponding phoneme transcriptions lies in the segmental structures in speech whose boundaries are unknown. To tackle this challenge, we propose **REBORN**, which trains a UASR system with unpaired speech and text data. REBORN contains a segmentation model and a phoneme prediction model. The segmentation model takes the speech feature as input and determines the boundaries of the segmental structures in the speech signal, and the phoneme prediction model predicts a phoneme for each segmental structure. We will use the terms *segment* and *segmental structure* interchangeably. In our paper, we do not use the term *segment* in the exact sense as in linguistics, where *segment* refers to discrete units that can be identified in the stream of speech [15]. We use *segment* to broadly refer to a span in the speech, which may be a meaningful unit (*e.g.*, a phone or word) or a span in the speech identified by the segmentation model.

The overall training process of REBORN is outlined in Figure 1. First, we initialize the phoneme prediction model using wav2vec-U (Section 3.3). Next, the training of REBORN iterates through two stages: **Stage 1** (Figure 1(a), Section 3.1): Training the segmentation model to learn a better segmentation until the segmentation model converges while fixing the phoneme prediction model. **Stage 2** (Figure 1(b), Section 3.2): Training the phoneme predictor based on the segment boundaries predicted by the segmentation model until the phoneme prediction model converges. The iterative process ceases when the UASR performance does not improve over the previous iteration. In UASR,

we cannot use PER as a validation metric to select the best model or determine when to stop training. We use an unsupervised evaluation metric to achieve it instead, detailed in Appendix C.4 and C.5.

### 3.1 Stage 1: Training the Segmentation Model

#### 3.1.1 Segmentation Model

Given an input speech signal, we use a self-supervised feature extractor model to extract the speech features $X = [x_1, \cdots, x_T]$ from the waveform. The segmentation model takes the speech features and predicts the boundary of the segmental structure in the speech features. For each feature $x_i$, the segmentation model assigns a binary value $\hat{b}_i$, indicating whether $x_i$ is the first frame of a segment. We split $X$ into segments based on $\hat{b} = \hat{b}_{1:T}$ and mean-pool the features within the same segment. The mean-pooled features $[s_1, \cdots, s_{T'}]$ are forwarded through the phoneme prediction model to obtain the phoneme prediction $[y_1, \cdots, y_{T'}]$ using greedy decoding. The resulting phoneme prediction will be de-duplicated into $[y'_1, \cdots, y'_M]$ by removing the same consecutive phoneme.

The segmentation model is a one-dimensional CNN, denoted by $\pi_\theta$ and parametrized by $\theta$. Given $[x_1, \cdots, x_T]$, $\pi_\theta$ predicts a probability $\pi_\theta[i]$, $\forall i \in [1, \cdots, T]$. Let $\hat{B}_i \sim \text{Bernoulli}(\pi_\theta[i])$ be a Bernoulli random variable representing whether $x_i$ is the beginning of a segment. $\hat{B}_i = 1$ means $x_i$ is the beginning of a segment; $\hat{B}_i = 0$ means $x_i$ is not the beginning of a segment. During training, we sample $\hat{b}_i$ from $\text{Bernoulli}(\pi_\theta[i])$; during inference, we take $\hat{b}_i = 1$ if $\pi_\theta[i] \geq 0.5$, otherwise $\hat{b}_i = 0$.

#### 3.1.2 Training the Segmentation Model with RL

To help the phoneme prediction model predict phonemes, we want to train the segmentation model to capture the segmental structure in speech. However, the optimal segmentation boundary is not available for training. Recognizing that the segmentation quality directly affects the phoneme prediction of the phoneme prediction model, we estimate the quality of the phoneme prediction to guide the segmentation model training. The phoneme prediction model is from the previous REBORN iteration or initialized from wav2vec-U in the first REBORN iteration, and it is fixed when training the segmentation model. Given that the feature segmentation based on boundary decision is inherently non-differentiable, we leverage RL to train the segmentation model. While related works may employ techniques like soft monotonic alignment or straight-through estimator to approximate gradients [7, 60], we find that RL provides a natural and suitable approach for our scenario, where the estimated quality of the phoneme sequence across different segmentations serves as the RL reward.

We train the segmentation model using the policy gradient method [57] based on the REINFORCE algorithm [63]. For each utterance, we calculate an utterance-wise reward $R$ (defined in Eq. 4 in the next subsection) from the de-duplicated phoneme prediction to train the segmentation model. Based on the policy gradient method, the segmentation model $\pi_\theta$ is optimized using the following gradient: $\mathbb{E}_{\hat{b} \sim \pi_\theta} \left[ \nabla_\theta \log \pi_\theta(\hat{b}|X) R \right]$, where $\mathbb{E}$ is taken over $\hat{b}$ sampled from $\pi_\theta$ and approximated with the mean of a batch of training sequences.

#### 3.1.3 Reward Designs

Given an utterance in the training set, the utterance-wise reward $R$ is the weighted sum of **perplexity difference reward** $R_{\text{ppl}}$, **edit-distance reward** $R_{\text{edit}}$, and **length difference reward** $R_{\text{len}}$.

We introduce some notations: Given an utterance, we use the currently trained segmentation model $\pi_\theta$ for segmentation and the phoneme prediction model trained in the last REBORN iteration to obtain the de-duplicated phoneme prediction sequence $Y'_\theta$. For the same utterance, we use the segmentation model from the **previous** REBORN iteration $\pi_{\theta-1}$ for features segmentation and the same phoneme prediction model to obtain another de-duplicated phoneme prediction sequence, denoted as $Y'_{\theta-1}$. In the first REBORN iteration, $Y'_{\theta-1}$ is the de-duplicated phoneme prediction from wav2vec-U.

**Perplexity Difference Reward.** The perplexity difference reward is designed to favor phoneme segmentation better than the segmentation learned in the previous iteration. Intuitively, a better phoneme segmentation prediction $\hat{b}_{1:T}$ should yield a more reasonable phoneme prediction $y'_{1:M}$. We use perplexity (PPL), the negative likelihood of a phoneme sequence scored by a phoneme language model (LM), to evaluate how reasonable a phoneme sequence is. Perplexity measures how likely a

phoneme sequence can be observed in the real world; a phoneme sequence with a lower perplexity means it is more likely to be observed in real-world phoneme datasets. We use a 4-gram phoneme LM trained on the phonemicized unlabeled text corpora. To guide the segmentation model to generate a segmentation with a better phoneme prediction than the phoneme predictions obtained in the previous iteration, we define **perplexity difference reward** as follows:

$$R_{\text{ppl}} = \text{PPL}_{\theta-1} - \text{PPL}_\theta, \tag{1}$$

where $\text{PPL}_{\theta-1}$ is the perplexity of $Y'_{\theta-1}$ and $\text{PPL}_\theta$ is the perplexity of $Y'_\theta$. The perplexity difference reward guides the segmentation model to produce segmentation that results in a phoneme sequence with lower perplexity comparing with the previous iteration.

When only using $R_{\text{ppl}}$, the segmentation model learns some segmentation method that leads to phoneme predictions with lower perplexity but does not correspond to better phoneme prediction results. To prevent such undesirable behaviors, we design two regularization rewards, the edit distance reward and length difference reward, to ensure that the policy learned by the segmentation model does not drastically alter the phoneme predictions between iterations. The two regularization rewards punish the segmentation model if $Y'_\theta$ is too different from $Y'_{\theta-1}$. Crucially, regularization works only if the $Y'_{\theta-1}$ in the first iteration is good enough. As previously mentioned, we ensure this by using the predictions from a trained wav2vec-U as $Y'_{\theta-1}$ in the first iteration.

**Edit distance reward.** We use Levenshtein distance $d_{\text{Lev}}$ as the edit distance to calculate the difference between $Y'_{\theta-1}$ and $Y'_\theta$. We take the normalized negative edit distance as the reward, which is defined in Eq. 2

**Length difference reward.** We use the length difference reward $R_{\text{len}}$ to guide the segmentation model to predict segmentation such that the length of $Y'_\theta$ does not differ significantly from the length of $Y'_{\theta-1}$. The length difference reward for an utterance is defined in Eq. 3, where $|Y'|$ is the length of $Y'$.

$$R_{\text{edit}} = -\frac{d_{\text{Lev}}(Y'_{\theta-1}, Y'_\theta)}{|Y'_{\theta-1}|}, \qquad (2) \qquad R_{\text{len}} = 1 - \frac{\left||Y'_\theta| - |Y'_{\theta-1}|\right|}{|Y'_{\theta-1}|} \qquad (3)$$

The final utterance-wise reward $R$ is the weighted sum of $R_{\text{ppl}}$, $R_{\text{edit}}$, and $R_{\text{len}}$.

$$R = c_{\text{ppl}} \cdot R_{\text{ppl}} + c_{\text{edit}} \cdot R_{\text{edit}} + c_{\text{len}} \cdot R_{\text{len}}, \tag{4}$$

where $c_{\text{ppl}}$, $c_{\text{edit}}$ and $c_{\text{len}}$ are the weighting coefficients. During training, $R_{\text{ppl}}$, $R_{\text{edit}}$ and $R_{\text{len}}$ are normalized within each batch. Appendix C.5 discusses how the coefficients are determined.

### 3.1.4 Initializing $\pi_\theta$ with Behavior Cloning

Before training the segmentation model $\pi_\theta$ with RL, we initialize it with behavior cloning (BC) [47, 48]. BC uses the supervised objective to train the segmentation model to predict the boundaries in speech features. Given speech features $x_{1:T}$, the segmentation model is trained to predict the 0/1 boundary $\hat{b}_i$ by using some boundaries $b_i$ as the target. In REBORN, the target boundary for BC is the *merged* boundary (Section 3.1.5) obtained using $\pi_{\theta-1}$, the segmentation model learned in the previous iteration. We then frame BC as a binary classification task and optimize it with the cross-entropy loss. Note that in the first REBORN iteration, we use the $k$-means-based boundary from wav2vec-U as the BC target.

### 3.1.5 Boundary Merging

After training the segmentation model until convergence, we use the segmentation model to predict the boundaries of the whole dataset and perform boundary merging. The process uses the phoneme prediction model trained in the previous iteration to refine the boundary predictions $\hat{b}_i$. Even if some consecutive speech features are split into two segments by the segmentation model, the mean-pooled features of the two segments may yield the same phoneme prediction. In this case, the two segments will be merged into one segment. An illustration of boundary merging is shown in Figure 1(c). Boundary merging differs from phoneme sequence de-duplication: boundary merging modifies the segmentation prediction $\hat{b}_i$, while de-duplication modifies the phoneme sequence.

## 3.2 Stage 2: Training the Phoneme Prediction Model

The phoneme predictor takes the mean-pooled features of segmental structures $S = [s_1, s_2, \cdots, s_{T'}]$ and predicts a phoneme sequence $Y = [y_1, y_2, \cdots, y_{T'}]$. $S$ are obtained by mean-pooling the features $X$ in the same segmental structure based on $[\hat{b}'_1, \cdots, \hat{b}'_T]$, where $\hat{b}'_i$ is the *merged* boundaries. We find that it is effective to perform boundary merging to stabilize the training in this stage.

After obtaining the merged phoneme boundaries, we train the phoneme predictor model using GAN training. The phoneme prediction model is the generator in GAN training. The generator aims to output phoneme predictions that look like real phoneme sequences to fool the discriminator. The discriminator takes in a phoneme sequence, which can be the output of the generator or a phoneme sequence from the unpaired text corpora, and the goal of the discriminator is to distinguish whether the input phoneme sequences are outputs of the generator. The generator and discriminator are updated using the loss in GAN training. In this stage, the parameters of the segmentation model are not updated, and the generator is initialized from the previous iteration. We discuss the effect of applying boundary merging and parameter initialization in Appendix A.1

### 3.3 Initialization of REBORN

In Stage 1, the segmentation model depends on the phoneme prediction when calculating the rewards. As a result, REBORN cannot work without properly initializing the phoneme prediction model. We use wav2vec-U to train a phoneme prediction model and use it as the initialization for the phoneme prediction model in REBORN. We briefly introduce wav2vec-U in Appendix B.

## 4 Experiment Setup

### 4.1 Dataset

We use three datasets commonly used in ASR to evaluate the performance of REBORN.

**LibriSpeech** [45] is an English speech recognition corpora that contains 960 hours of training data. Following EURO [18], we use 100 hours of audio from the train-clean-100 set as the unlabeled speech data. The unlabeled text data is derived from the remaining 860 hours, which does not overlap with the transcription of the unlabeled speech data.

**TIMIT** [19] is another English speech recognition with the human-labeled phone boundary. We follow the *matched* setting, which is more broadly used [4, 18, 33], where the speech and text data come from the same set of utterances.

**Multilingual LibriSpeech (MLS)** [50] is an ASR dataset including *German (de)*, *Dutch (nl)*, *French (fr)*, *Spanish (es)*, *Italian (it)*, and *Portuguese (pt)*. Following [4], we randomly sample and use 100 hours of speech data for each language and use the LM data provided by the dataset as unpaired text.

### 4.2 Evaluation Metrics

We use phoneme error rate (PER) and word error rate (WER) to evaluate the ASR performance. If not specified, we use greedy decoding to obtain phoneme-level results. For decoding word-level outputs, we perform WFST decoding [38, 39] using PyKaldi [9]. We leave the details in Appendix C.7.

### 4.3 Implementation Details

For the English datasets, we use wav2vec 2.0 [3] to extract speech features from speech signals; for MLS, we use XLSR-53 [14] as the feature extractor. Both of them can be found in fairseq [44]. The 4-gram phoneme LM is derived using KenLM [22]. We describe more details about the data preparation and LM formulation in Appendix C.1, which basically follows wav2vec-U [4]. All the trainable models, including the segmentation model, the phoneme prediction model, and the discriminator in GAN training, are composed of one-dimensional CNN layers, as detailed in Appendix C.3. For LibriSpeech and TIMIT, we train REBORN for two iterations. For MLS, we only train one iteration since we do not find the performance to improve in the second iteration.

Table 1: PER/WER on LibriSpeech using 100 hours speech data. †: Our reproduction. (wav2vec-U and wav2vec-U 2.0 only report results of using 960 hours of unlabeled speech). HMM ST indicates HMM self-training (Appendix C.2).

| Approach | PER/WER (%) ↓ | | | |
|---|---|---|---|---|
| | dev-clean | dev-other | test-clean | test-other |
| **WITH ORACLE BOUNDARY** | | | | |
| Train from oracle | 6.3/12.8 | 9.7/16.3 | 6.4/12.7 | 10.0/16.8 |
| **BASELINE** | | | | |
| EURO (HuBERT) | 15.2/23.1 | 20.7/29.3 | 15.1/22.8 | 21.1/29.8 |
| wav2vec-U† | 19.3/20.4 | 22.9/25.6 | 19.3/21.0 | 23.2/25.8 |
| wav2vec-U 2.0† | 12.2/17.2 | 16.3/21.7 | 12.6/17.7 | 16.3/22.2 |
| wav2vec-U 2.0 + HMM ST† | 10.0/15.4 | 13.1/19.0 | 10.3/16.0 | 13.1/19.6 |
| **OUR METHOD** | | | | |
| REBORN | **8.3/12.5** | **11.9/17.6** | **8.9/13.1** | **12.5/18.7** |
| REBORN + HMM ST | **5.2/9.3** | **8.5/13.5** | **5.4/9.6** | **8.5/13.7** |

Table 2: PER results on TIMIT. The cross-mark (✗) in the greedy-decoding column indicates that an additional LM (4-gram) is used during decoding. REBORN reaches the best performance with no LM used for decoding, showing that REBORN can benefit from the external LM via RL.

| Approach | Greedy decoding | PER (%) | | |
|---|---|---|---|---|
| | | core-dev | core-test | all-test |
| (a) Train from oracle | ✓ | 9.1 | 10.3 | 9.2 |
| (b) wav2vec-U (reproduced) | ✓ | 20.2 | 22.2 | 20.3 |
| (c) wav2vec-U+WFST | ✗ | 17.1 | 17.8 | 16.8 |
| (d) EURO (wav2vec 2.0) | ✗ | 18.5 | 19.8 | - |
| (e) EURO (WavLM) | ✗ | 14.3 | 14.6 | - |
| (f) REBORN | ✓ | **12.4** | **13.5** | **12.4** |

# 5 Results

## 5.1 Main Results

We show the PER and WER of LibriSpeech, TIMIT, and MLS in Table 1, Table 2, and Table 3. We compare REBORN with several prior UASR works: wav2vec-U [4], wav2vec-U 2.0 [30], and EURO [18]. We have the following observations:

**REBORN significantly outperforms all prior methods on LibriSpeech.** Our experiment on LibriSpeech follows Gao et al. [18] to use the 100-hour training split of LibriSpeech as the unlabeled speech data. Compared with all the baselines in Table 1, REBORN achieves the lowest PER and WER. wav2vec-U and EURO both use hand-crafted rules to obtain the segmental boundaries, while wav2vec-U 2.0 removes the feature segmentation steps. The superior performance of REBORN illustrates the importance of learning the segmental boundaries tailored for the phoneme prediction model. Notably, without using HMM self-training, REBORN already has PER/WER lower than wav2vec-U 2.0 with HMM self-training, which is the prior state-of-the-art (SoTA) method on LibriSpeech. In Appendix A.4, we further apply the REBORN pipeline with speech foundation models other than wav2vec 2.0. We find that REBORN yields notable performance improvement when using HuBERT [23] or WavLM [12] as the feature extractor, illustrating the generalizability of our method.

**Self-training further improves PER/WER of REBORN.** Self-training uses the phoneme predictor's prediction as the pseudo-label and trains a new phoneme prediction model using the pseudo-label as the training data. It is commonly used in UASR to boost the performance [4, 30]. In Table 1, we show that REBORN can be integrated with Hidden Markov Models (HMM) self-training (Appendix C.2) to further lower the PER/WER. We reach a new SoTA on LibriSpeech under the setting of 100-hour speech data, outperforming the prior SoTA in PER and WER by 5% and 6%, respectively. Surprisingly, REBORN with HMM self-training outperforms training wav2vec-U with the oracle boundary. This shows that REBORN can be combined with self-training to improve performance effectively. Due to limited computation resources, we only conduct self-training on LibriSpeech.

**REBORN outperforms all prior UASR methods on TIMIT.** Table 2 shows the PER of TIMIT, and we again find that REBORN achieves the lowest PER compared with all prior UASR methods. This indicates that REBORN not only works on large datasets like LibriSpeech but also works on small datasets like TIMIT, which only contains about 3 hours of audio data for training. Even using greedy decoding only, REBORN outperforms the prior best-performing UASR model (row (e)), which relies on prefix decoding. EURO shows that replacing the feature extractor with WavLM [12] (row (e)) outperforms using wav2vec 2.0 (row (d)) as the speech feature extractor. Since REBORN using wav2vec 2.0 already outperforms EURO with WavLM, we leave changing the feature extractor in REBORN on TIMIT as future work.

Table 3: WER on MLS. †: Results from Baevski et al. [4]. ∗: Our reproduction of wav2vec-U, used as the initialization of the phoneme prediction model in REBORN. ‡: Results from Liu et al. [30].

| Approach | WER (%) ↓ | | | | | | |
|---|---|---|---|---|---|---|---|
| | de | nl | fr | es | it | pt | Avg. |
| wav2vec-U† | 32.5 | 40.2 | 39.8 | 33.3 | 58.1 | 59.8 | 44.0 |
| wav2vec-U∗ | 33.9 | 38.1 | 37.7 | 33.1 | 51.8 | 59.4 | 42.3 |
| wav2vec-U 2.0‡ | 23.5 | 35.1 | 35.7 | 25.8 | 46.9 | **48.5** | 35.9 |
| REBORN | **20.9** | **26.9** | **28.2** | **24.7** | **39.9** | 51.5 | **32.0** |

Table 4: Boundary evaluation results of different segmentation methods on LibriSpeech test-clean split. The second-last column (Freq.) is the number of segments per second. All the methods share the same phoneme prediction model trained with the $k$-means-based segmentation of wav2vec-U.

| Boundary method | Prec. | Rec. | F1 | Freq. | PER% |
|---|---|---|---|---|---|
| (a) Oracle | 1.0 | 1.0 | 1.0 | 11.89 | 13.8 |
| (b) $k$-means-based | 0.64 | 0.77 | 0.70 | 14.27 | 19.3 |
| (c) Strgar and Harwath [56] | **0.75** | 0.75 | **0.75** | 12.26 | 23.2 |
| (d) REBORN | 0.57 | **0.78** | 0.65 | 16.29 | **12.9** |

**REBORN performs well on languages other than English.** The results on MLS are presented in Table 3. We find out that a single iteration is enough for performance convergence on MLS. Recall that REBORN is initialized from wav2vec-U, but REBORN outperforms wav2vec-U by a large margin in all six languages. This large performance gap underlines the importance of learning the boundaries of the segmental boundaries. Moreover, REBORN outperforms wav2vec-U 2.0 except for Portuguese, and the average WER on the six languages is 3.9% lower than wav2vec-U 2.0.

## 5.2 Boundary Analysis

The core of REBORN is the segmentation model that learns the segmental structure's boundaries. In this section, we look at how different segmental boundaries affect the performance of phoneme prediction. We compare the boundaries obtained by four methods: (a) the oracle phoneme boundaries obtained by forced alignment [36], which requires paired speech-text data to learn the alignment; (b) the $k$-means-based segmentation boundary in wav2vec-U; (c) the phoneme boundary obtained by the SoTA unsupervised phoneme segmentation method [56] on LibriSpeech; (d) the boundaries learned by the segmentation model of REBORN in the first iteration **before** boundary merging.

First, we focus on the PER in Table 4 when using different segmentation methods. The PER is obtained by taking the four different segmented features to the identical phoneme prediction model: the phoneme prediction model trained using the $k$-means-based segmentation of wav2vec-U. Note that the phoneme prediction model here is **unsupervised** and **non-ideal**. We find that simply replacing the $k$-means-based boundary with the oracle phoneme boundary reduces the PER by 5.5%, showing the imperfect hand-crafted $k$-means-based segmentation is the bottleneck of UASR. Next, even when using the boundary predicted by the SoTA unsupervised phoneme segmentation [56], the PER does not reduce. Conversely, the boundary learned by REBORN achieves the lowest PER. This is because the boundary learned by REBORN is directly tailored for the phoneme prediction model.

Next, we discuss the phoneme boundary results in Table 4. The phoneme boundary results is evaluated with boundary precision, recall, and F1 with a 20ms tolerance window, following the *harsh* scheme from Strgar and Harwath [56]. We leave the discussion about the different schemes for boundary evaluation in Appendix E. We also report the average number of segments per second (Freq. in Table 4). Interestingly, the boundary F1s of the oracle boundary (row (a)) and Strgar and Harwath [56] (row (c)) are much higher than the boundary F1 of REBORN (row (d)), but the PER of REBORN is much better. This is because REBORN's segmentation model predicts more segments than the number of phonemes in the utterance (before boundary merging), indicating that it learns some segmental structures smaller than the phones. This can be seen from the lower precision (0.57) and higher frequency (16.29) compared to oracle boundaries. However, segmenting the speech feature into smaller units is not problematic because even if the consecutive speech features of the same phonemes are split into two segments by the segmentation model, as long as they have the same phoneme prediction, the duplicated phoneme prediction will be removed in the de-duplication step, and thus not affect the PER.

Table 5: We compare different reward functions and the effect of BC initialization with first iteration results on LibriSpeech test-clean. Row (c) is REBORN.

| Ablation | PER (%) | | LM PPL |
| | Stage 1 | Stage 2 | (Stage 1) |
|---|---|---|---|
| (a). $R_{ppl}$ | 14.7 | 12.9 | 10.0 |
| (b). $R_{ppl} + R_{edit}$ | 13.8 | 12.1 | 10.8 |
| (c). $R_{ppl} + R_{edit} + R_{len}$ | **12.9** | **11.9** | 11.2 |
| (d). REBORN w/o BC | 14.9 | 14.2 | 13.8 |

Figure 2: PER across training epochs on the test-clean split of LibriSpeech. BC pretraining speeds up convergence and raises performance.

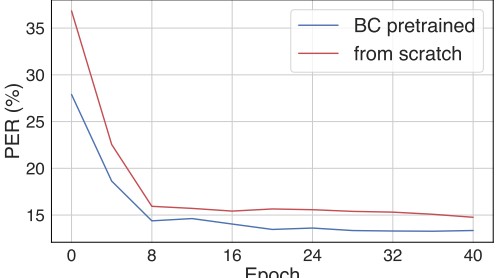

## 5.3 Ablation Study

### 5.3.1 Reward Design and Behavior Cloning

We verify the effectiveness of our three proposed reward functions and behavior cloning initialization in Table 5. While the perplexity difference reward alone is sufficient to guide the segmentation model, adding edit distance and length difference rewards improves performance further. As previously mentioned, using only the perplexity difference reward will make the segmentation model predict segments overly focusing on lowering PPL. And a phoneme sequence with a lower perplexity may not always be a better transcription. To deal with such overfitting problem, we design the two regularization rewards, namely, $R_{edit}$ and $R_{len}$, to make the model learn to segment for lower perplexity while grounded on a reasonable transcription from the previous iteration. Our results in Table 5 further evidence the effectiveness of the two rewards. Additionally, we observe that removing BC leads to a decline in PER compared to using BC (see row (d) vs. row (c)). To deepen this analysis, we present the PER curve on the LibriSpeech *test-clean* set with and without BC. The results in Figure 2 indicate that BC helps enhance performance and accelerate convergence.

### 5.3.2 Iterative Training

In Figure 3, we show the PER on LibriSpeech test-clean split during the iterative training process of REBORN. We observe that after training the segmentation model in the first iteration, the PER drastically drops by 6.4% compared with wav2vec-U, which is used to initialize the phoneme prediction model in iteration one. This shows that the quality of the segmental structure's boundaries is important to the performance of UASR. Afterward, the PER of REBORN decreases steadily across iterative training, showing that training the segmentation model and the phoneme prediction model is critical to improving the performance of REBORN. We find that the PER does not drop significantly after the fifth iteration. Last but not least, we provide more evidence in Appendix A.3 to showcase that the phoneme predictors are iteratively polished.

Figure 3: PER of each stage during REBORN's two-stage iterative training on the test-clean split of LibriSpeech. St.: stage; w2vu: wav2vec-U.

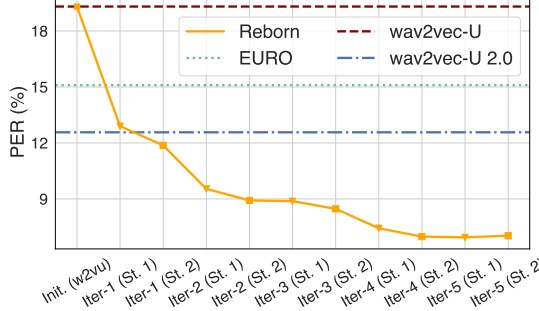

Table 6: The boundary evaluation results on TIMIT. REBORN achieves better boundary evaluation scores than the original $k$-means-based method, and is comparable with Strgar and Harwath [56]'s after boundary merging. ∗: from literature. All the metrics are calculated on the *test-all* split.

| Boundary method | Prec. | Rec. | F1 | R-val. | PER% |
|---|---|---|---|---|---|
| $k$-means-based | 0.62 | 0.75 | 0.68 | 0.68 | 20.3 |
| Strgar and Harwath [56]∗ | 0.85 | 0.79 | 0.82 | 0.84 | - |
| REBORN | 0.61 | 0.83 | 0.71 | 0.62 | 12.4 |
| + boundary merging | 0.80 | 0.78 | 0.79 | 0.82 | - |

### 5.3.3 Boundary Evaluation on TIMIT

In Section 5.2, we demonstrate that REBORN segmentation delivers fine-grained segments, achieving the highest performance gain among segmentation methods given the same non-ideal phoneme prediction model. Here, we extend our boundary evaluation results to TIMIT, a smaller dataset with human-annotated phone boundaries, to provide more comprehensive boundary insights. Table 6 shows that the initial boundaries learned by REBORN, optimized explicitly for the phoneme prediction model, already achieve a high recall. With boundary merging, *i.e.*, consecutive segments with the same phoneme prediction are merged as illustrated in Figure 1-(c), REBORN achieves results close to Strgar and Harwath [56]'s. In line with the previous context, we emphasize that REBORN achieves substantial performance improvement in PER by tailoring segmentation to the non-ideal unsupervised phoneme prediction model, rather than solely aiming for higher boundary evaluation scores. This represents a critical and intriguing finding in our work.

## 6  Discussion

In unsupervised ASR, learning the mapping between the segmental structures in speech and their corresponding transcription is challenging, especially without paired data. To tackle this, we propose REBORN, an iterative training method for unsupervised ASR. REBORN iteratively trains the segmentation model and the phoneme prediction model to improve the ASR performance. The segmentation model is trained using RL with carefully designed rewards to guide the segmentation model to find the boundaries of the segmental structures in speech signals. Extensive experiments on three datasets spanning seven languages show that REBORN outperforms prior best-performing unsupervised ASR models in all datasets except one language in MLS. We conduct comprehensive ablation studies to show that each component in REBORN is critical to the performance. We also explain the effectiveness of REBORN by analyzing its segmentation patterns and find that REBORN tends to produce segmental structures smaller than phones, which helps the generators predict the phoneme transcription better.

**Limitations.** Recent advancements in unsupervised ASR are still in the developmental stages, and we have not yet implemented REBORN in more realistic scenarios, such as low-resource languages or noisy environments. Additionally, the iterative training paradigm may amplify any existing biases in the dataset, an aspect that remains unexplored in this study. Furthermore, if the phoneme prediction model is poorly initialized, REBORN may not be able to provide huge performance improvements. In this work, lexicons are required to perform phonemicization on the unpaired text. Consequently, learning the unsupervised ASR system directly from speech to word-level tokens remains an open problem, where recent works aim to tackle [30, 41].

**Broader impacts.** Unsupervised automatic speech recognition (UASR) - predicting the textual transcriptions for the given speech signals using only unpaired speech and text data - is highly attractive. The existence of thousands of low-resourced languages spoken globally is the major reason. This paper presents REBORN, a novel unsupervised ASR training algorithm. We foresee that REBORN has great potential to make low-resource languages more accessible to train ASR systems, making speech technology more accessible and boundaryless.

## Acknowledgment

We thank the National Center for High-performance Computing (NCHC) of the National Applied Research Laboratories (NARLabs) in Taiwan for providing computational and storage resources. Shao-Hua Sun was supported by the Yushan Fellow Program by the Ministry of Education, Taiwan. We also extend our gratitude to Alexei Baevski, Wei-Ning Hsu, and Alexander H. Liu, authors of wav2vec-U and wav2vec-U 2.0, for providing valuable insights into their work.

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

# Appendix

## A  Additional Results

### A.1  Stability Analysis

In unsupervised ASR, stability has always been one of the major concerns since the unlabeled scenario makes the problem difficult and the training unstable. In Table 7, we demonstrate that REBORN is stable by calculating the mean PER and the corresponding standard deviations, as well as the converged rate. We introduce that *a converged experimental run* is categorized by its result yielding PER $< 40\%$, following Baevski et al. [4]. As represented in the table, REBORN generates significantly better performance while being stable with a 100% converged rate in both stages.

Next, we focus on the ablations of REBORN stage 2 GAN training in Table 7. When the boundary merging is not applied, the GAN training on the original REBORN-segmented features is still better than the initial stage. However, the average PER and the standard deviation are larger. The observation indicates that the training is slightly unstable, but does not diverge too much since it is still fully converged. On the other hand, when we train the next-iteration GAN with completely random initialized parameters, the performance degrades even if we train the model for more steps. These two ablations directly demonstrate the effectiveness of adopting boundary merging and parameter initialization from the previous iteration, while the GAN training can still work on the original boundary or with randomly initialized parameters.

Furthermore, we find out that the instability might be highly correlated to the boundary frequency (the last column in Table 7), and this observation also corresponds with the prior works [4, 30]. Just as we performed in Table 1, we show the topline result of training GAN from the oracle boundary in the last row. We contribute the good result to the disappearance of the length mismatch issue between the segmented feature and its textual transcription. In the INITAIL STAGE, when we use only the $k$-means for segmentation instead of the two-stage $k$-means + adjacent segmentation (or "$k$-means-based segmentation" in the main text) in wav2vec-U, the length mismatch issue arises and we can not obtain reasonable results.

As for REBORN, the boundary without merging has a higher frequency compared with the wav2vec-U segmentation. However, using the original boundary for GAN training can still yield fully converged results with lower PER. We contribute this result to the improved quality of REBORN boundary, showing the importance and effectiveness of our segmentation learning via RL.

Table 7: The table for stability analysis. Each method is trained for 5 runs on LibriSpeech and evaluated on the *test-clean* split. REBORN performs steadily well while being fully converged in both of the stages. The ablations show the effectiveness of applying boundary merging and parameter initialization from the previous iteration for the stage 2 GAN training. The abbreviation "adj." indicates adjacent pooling, which is the 2nd stage of pooling in wav2vec-U (see Appendix B). The "Freq." in the last column is the number of boundaries per second.

| Ablation | Type | mean PER $\pm$ std (%) | %-converged ($\uparrow$) | Freq. (Hz) |
|---|---|---|---|---|
| **INITIAL STAGE** | | | | |
| wav2vec-U ($k$-means-only) | GAN | $> 100$ | 0% | 28.5 |
| wav2vec-U ($k$-means + adj.) | GAN | $20.1 \pm 0.6$ | 100% | 14.3 |
| **ITERATION 1** | | | | |
| REBORN stage 1 | RL | $12.9 \pm 0.7$ | 100% | 16.3 |
| REBORN stage 2 | GAN | $12.0 \pm 0.1$ | 100% | 11.6 |
| w/o boundary merging | GAN | $13.9 \pm 2.1$ | 100% | 16.3 |
| w/ random initialization | GAN | $14.1 \pm 0.4$ | 100% | 11.6 |
| **TOPLINE** | | | | |
| Train from oracle boundary | GAN | $6.64 \pm 0.2$ | 100% | 11.9 |

Table 8: Ablation study results of various configurations on the unsupervised metric assessed using validation set and phoneme error rates evaluated on the LibriSpeech dataset.

| Configuration | Stage 1 | | | Stage 2 | | |
|---|---|---|---|---|---|---|
| | Unsupervised Metric ($\downarrow$) | Dev-clean PER (%) | Test-clean PER (%) | Unsupervised Metric ($\downarrow$) | Dev-clean PER (%) | Test-clean PER (%) |
| $R_{\text{ppl}}$ | 256621.6 | 14.0 | 14.7 | 206312.79 | 12.2 | 12.9 |
| $R_{\text{ppl}} + R_{\text{edit}}$ | 251476.6 | 13.3 | 13.8 | 198447.93 | **11.7** | 12.1 |
| $R_{\text{ppl}} + R_{\text{edit}} + R_{\text{len}}$ | **243419.4** | **12.7** | **12.9** | **191043.37** | **11.7** | **11.9** |
| REBORN w/o BC | 252302.3 | 14.8 | 14.9 | 237658.23 | 14.1 | 14.2 |

## A.2 Reward Design Details

When training reinforcement learning, various methods have demonstrated that incorporating regularization terms can enhance model training stability [54, 55]. In response to this insight, we have integrated both edit-distance and length rewards as forms of regularization in our training regime. To identify the most effective reward configuration, we employed an unsupervised metric (Eq. 10) for selection. Detailed unsupervised metric scores and further evaluation results of our reward configurations are presented in Table 8.

Our findings, as shown in Table 5, indicate that using perplexity (PPL) alone as a reward yields the lowest perplexity scores upon model convergence. However, this approach also results in higher scores on the unsupervised metric, implying a decline in model performance. Further analysis reveals that the PER increases when the PPL reward is the sole metric. This suggests that, without regularization, the model tends to minimize perplexity by producing more lengthy and less accurate transcriptions, which do not align well with the expected outputs.

## A.3 Iteratively Polished Phoneme Predictors

In REBORN, we introduce an iterative paradigm that can gradually polish the segmentation model and the phoneme predictor in turn. In Table 9, we wish to provide more evidence to directly demonstrate that the phoneme predictors are actually improved through the iterations. We carry this out by feeding the identical oracle-segmented features to the phoneme predictors in different stages. Given the same ideally-segmented features, a better phoneme predictor should yield a better performance as well. In Table 9, we show that the phoneme predictor in the initial stage gives 13.8% PER when evaluating on the oracle-segmented features. REBORN's iterative learning paradigm gradually improves the phoneme predictor, and the performance finally achieves the best in the 4th iteration by 7.4% PER. The relative performance gain compared with the initial stage is over 45%, as presented in the table. The result is even 1% close to the topline listed in the bottom row, indicating the effectiveness of REBORN's iterative training.

Table 9: We demonstrate that the REBORN phoneme predictors are gradually improved though our iterative training. Each of the phoneme predictors takes the same **oracle-segmented** features as input.

| Phoneme Predictor | PER (%) (evaluate on oracle) | Relative Gain ($\uparrow$) |
|---|---|---|
| **INITIAL STAGE** | | |
| wav2vec-U | 13.8 | 0% |
| **REBORN** | | |
| Iteration 1 | 10.9 | 21% |
| Iteration 2 | 9.9 | 28% |
| Iteration 3 | 7.6 | 45% |
| Iteration 4 | **7.4** | **46%** |
| Iteration 5 | 8.7 | 37% |
| **TOPLINE** | | |
| Train from oracle | 6.4 | 54% |

Table 10: We implement REBORN across different speech foundation models on LibriSpeech. The results are evaluated on *test-clean*. REBORN has exhibited strong generalizability by providing substantial performance improvements across different speech foundation models. We extract the 15th layer representations from HuBERT and WavLM following EURO [18].

| Feature extractor | PER on *test-clean* ($\downarrow$) | | | PER diff. |
| --- | --- | --- | --- | --- |
| | Initial stage | Iter.1-stage1 | Iter.1-stage2 | after Iter.1 |
| wav2vec 2.0 [3] | 19.3% | 12.9% | 11.9% | -7.4% |
| HuBERT [23] | 20.1% | 12.8% | 10.4% | -9.7% |
| WavLM [12] | 18.7% | 14.7% | 12.7% | -6.0% |

### A.4 Generalizability across Different Speech Foundation Models

We evaluate REBORN's generalizability across different speech foundation models in Table 10. Specifically, in Table 10, we replace the feature extractor from wav2vec 2.0 with HuBERT [23] or WavLM [12]. As presented in the table, REBORN can yield performance improvements regardless of which speech foundation model is used and how well the initial stage performed. The results further strengthen the robustness and the generalizability of our method. Due to the computational resource limitation, we only conduct the ablation for one iteration.

## B wav2vec-U

wav2vec-U [4] is a popular UASR framework whose variants [18, 30] have achieved SoTA performance on multiple UASR benchmarks. Following prior works [33], wav2vec-U also uses adversarial training to train the phoneme prediction model. The wav2vec-U framework takes the raw waveform as the input and predicts phoneme transcription via three stages: (1) the feature extraction stage, (2) the feature preprocessing stage, and (3) the phoneme prediction stage.

Stage (1) extracts speech features using a self-supervised speech foundation model, wav2vec 2.0 [3]. For a spoken utterance, we denote the extracted feature as $X = [x_1, x_2, ..., x_T] \in \mathbb{R}^{T \times d}$, where $T$ is the time dimension and $d$ is the feature dimension. Stage (2) is the feature preprocessing stage. The dimension of the extracted speech features is first reduced using PCA from $d = 1024$ to $d' = 512$. Next, $k$-**means clustering** and additional heuristics, namely, **adjacent pooling**, are adopted sequentially to define the boundary of the segmental structures in the speech features. The features in the same segmental structure are mean-pooled over the two stages into one feature embedding to reduce sequence length from $T$ to $T'$ to match phoneme sequences better. In the main text, we call this process "$k$-**means-based segmentation**". We denote the output from the feature preprocessing stage as $S = [s_1, s_2, ..., s_{T'}] \in \mathbb{R}^{T' \times d'}$. In Stage (3), the phoneme prediction model takes the pooled speech feature as input and predicts the phoneme sequence $Y = [y_1, y_2, ..., y_{T'}]$. Last, a de-duplication step removes the consecutive duplicate phoneme predictions from $Y$, resulting in a shortened phoneme sequence $Y' = [y'_1, y'_2, ..., y'_M]$ without consecutive duplicate phonemes.

During training, the generator $\mathcal{G}$ takes the preprocessed speech feature $S$ as input and generates the logits of $Y$. We can shorten the logits of $Y$ based on its argmax prediction, and the shortened logits (denoted as $G = \mathcal{G}(S)$) are then taken as input by the discriminator $\mathcal{D}$ for GAN training. The adversarial learning loss is defined as follows:

$$\mathcal{L}_{\text{GAN}} = \min_{\mathcal{G}} \max_{\mathcal{D}} \mathop{\mathbb{E}}_{Z \sim \mathcal{Z}}[\log(\mathcal{D}(Z))] + \mathop{\mathbb{E}}_{S \sim \mathcal{S}}[\log(1 - \mathcal{D}(\mathcal{G}(S)))] \tag{5}$$

, where $Z$ is a tokenized phoneme sequence sampled from the phonemicized text corpora $\mathcal{Z}$ and $S$ is the preprocessed speech feature sampled from the preprocessed speech corpora $\mathcal{S}$. Besides the min-max objective in the adversarial training, wav2vec-U [4] also adopts the three regularized objectives for better convergence and performance. First, the gradient penalty $\mathcal{L}_{\text{gp}}$ [21] is applied on the discriminator $\mathcal{D}$ to stabilize training, where the input of $\mathcal{D}$ is the linear combination of a generated

sample $G$ and a real sample $Z$.

$$\mathcal{L}_{\text{gp}} = \mathop{\mathbb{E}}_{G,Z,\alpha\sim U(0,1)} \left[ (\|\nabla\mathcal{D}(\alpha G + (1-\alpha)Z)\|_2 - 1)^2 \right] \tag{6}$$

Next, the smoothness penalty $\mathcal{L}_{\text{sp}}$ is computed across consecutive segments of a generated logit sequence, promoting the generation of closely aligned neighboring outputs by the model.

$$\mathcal{L}_{\text{sp}} = \sum_{(g_t, g_{t+1}) \in G} \|g_{t+1} - g_t\|^2 \tag{7}$$

Furthermore, the phoneme diversity loss $\mathcal{L}_{\text{pd}}$ is applied to encourage higher vocabulary usage of the generated sequences. As shown in Eq. 8, the entropy of the averaged softmax distribution of the generator outputs ($H_{\mathcal{G}}(\mathcal{G}(S))$) is maximized over the phoneme vocabulary across a batch B.

$$\mathcal{L}_{\text{pd}} = \frac{1}{|B|} \sum_{S \in B} -H_{\mathcal{G}}(\mathcal{G}(S)) \tag{8}$$

Finally, we describe the overall training objectives $\mathcal{L}$ by weighting with coefficients $\lambda$, $\gamma$, and $\eta$, respectively.

$$\mathcal{L} = \mathcal{L}_{\text{GAN}} - \lambda\mathcal{L}_{\text{gp}} + \gamma\mathcal{L}_{\text{sp}} + \eta\mathcal{L}_{\text{pd}} \tag{9}$$

In this work, we utilize the same training objective from wav2vec-U to perform GAN training (§ 3.2).

## C Implementation Details

### C.1 Data Preparation and Phoneme LM Formulation

Our data preparation process mainly follows wav2vec-U[2] for fair comparisons. Specifically, we use wav2vec 2.0 large pretrained on LibriLight [24] as the feature extractor in English[3]; and the multilingual version of wav2vec 2.0 (XLSR-53) for the other langauges[4]. Before the feature extraction stage, we attempt to remove most of the silence through an unsupervised voice activity detection (VAD) method [58][5]. However, the method is not completely perfect. To stabilize the GAN training, a special *<silence>* token is introduced to the text data. The *<silence>* token is inserted in the front and the end of each sentence, and also between words by a probability of 0.25 according to wav2vec-U. Then, we phonemicize all the words in the text data along with the previously inserted *<silence>* tokens for GAN training. The two off-the-shelf phonemizers are used to perform phonemicization. For LibriSpeech, we use the G2P phonemeizer [46], and the numerical stress makers are removed to reduce the phoneme set, which is shown to be beneficial for the performance [4, 18, 30]. As for Multilingual LibriSpeech, we use Phonemizer [6][6], a toolkit that supports a wide variety of languages other than English. The language-switching labels are removed as well as the phonemes appear less than 1000 times in the text data. For the Italian, we isolate the double consonant symbol in the lexicon, which is crucial for the initial stage to converge. Note that for TIMIT we do not adopt any of the above data preparations since it originally contains its phone inventories including silence. We use the 39-phone inventory which can be easily found in Kaldi [49] for training and evaluation[7], following prior UASR works [4, 18, 33].

After the data preparation, we directly use the phonemicized text with *<silence>* token to train the phoneme LM. More specifically, we utilize KenLM [22][8], to build the 4-gram LM mentioned

---

[2] https://github.com/facebookresearch/fairseq/tree/main/examples/wav2vec/unsupervised

[3] https://dl.fbaipublicfiles.com/fairseq/wav2vec/wav2vec_vox_new.pt

[4] https://dl.fbaipublicfiles.com/fairseq/wav2vec/xlsr_53_56k.pt

[5] https://github.com/zhenghuatan/rVAD

[6] https://github.com/bootphon/phonemizer

[7] https://github.com/kaldi-asr/kaldi/blob/master/egs/timit/s5/conf/phones.60-48-39.map

[8] https://github.com/kpu/kenlm

in 3.1.3. Since in the GAN training stage, we are actually matching the two distributions between the segmented speech features and phonemicized text with *<silence>*, we naturally build our external LM for RL segmentation learning from the same preprocessed text. The LM is also utilized for the unsupervised cross-validation described in C.5.

## C.2 Self-Training with Hidden Markov Models

Originated from semi-supervised learning, self-training aims at providing a trivial approach to utilize the unpaired or unlabeled data effectively [25, 35, 59]. More specifically, given a supervised ASR system $M_{ASR}$ learned from the labeled speech-text data $D_{labeled}$, we can use $M_{ASR}$ to generate pseudo labels of a significant but unlabeled speech data $D_{unlabeled}$. Now, we can utilize the speech data along with the pseudo labels to perform supervised learning or re-tuning [59] to obtain a new ASR system $M'_{ASR}$. Generally, given that the pseudo label is good enough, we may expect that $M'_{ASR}$ performs better than $M_{ASR}$.

In the unsupervised setup, the core concept of self-training is suited as well. By simply using an unsupervised ASR system $M_{UASR}$ to perform pseudo labeling, we can train a new ASR model $M'_{UASR}$ by the supervised objective. Since no labeled data is used (we are just using the pseudo labels), the new ASR system does not violate the unsupervised setup. More specifically, we use Hidden Markov Models (HMMs; Bosch et al. [8])as the backbone of the new ASR system. This can be dated back to [11] and is also adopted by recent UASR [4, 30]. Furthermore, to give fair comparisons and increase reproducibility, we follow the publicly available code provided by wav2vec-U for HMM self-training[9]. Our results in Table 1 indicate that with the high-quality pseudo labels generated by REBORN, the new ASR model based on HMMs performs better than the original UASR system and is the best among all the other methods.

## C.3 Model Architecture

We parametrize the segmentation model $\pi_\theta$ using a two-layer 1-dimensional convolutional neural network (CNN). This choice of architecture allows us to efficiently extract relevant information from adjacent speech features, making it particularly well-suited for our segmentation task. The two convolutional layers in our CNN have different kernel sizes, namely 7 and 3. This is a lightweight model, making it computationally efficient for applications. The phoneme prediction model (generator) and the discriminator in GAN training are parametrized by a single-layer 1-dimensional CNN and a two-layer 1-dimensional CNN, respectively, following wav2vec-U. Note that the phoneme prediction model generates the phoneme sequences non-autoregressively due to its architecture.

## C.4 Hyperparameter Search

We search the optimal hyperparameters of our phoneme predictor and segmentation models with unsupervised validation metrics. For the phoneme predictor, we directly adopt the search space of the hyperparameters and the unsupervised validation metric from wav2vec-U [4].We search the hyperparameters indicated as $\lambda$, $\gamma$, and $\eta$ in Appendix B only during the initialization stage of the phoneme predictor (§ 3.3). We directly adopt the same hyperparameters in the following iterations of GAN training. As for the segmentation model, we take the idea from Baevski et al. [4] and derive a similar unsupervised metric.

## C.5 Unsupervised Validation Metric for Segmentation Model Selection

Each segmentation model, denoted as $\pi_\theta$, undergoes unsupervised performance evaluation by predicting the transcriptions of the validation set. The phoneme predictions generated by the model are processed to form a de-duplicated set, denoted as $Y'_\theta = [y'_{1,\theta}, \cdots, y'_{M,\theta}]$, where consecutive duplicate phonemes are removed.

The unsupervised metric places significant emphasis on two key aspects: Language Model (LM) negative log-likelihood (NLL) and vocabulary usage. Vocabulary usage is calculated as $U(Y'_\theta) = \frac{1}{|V|} \sum_{v \in V} \mathbb{I}(v \in Y'_\theta)$, where $V$ denotes the entire vocabulary set. The validation metric seeks to

---

[9]`https://github.com/facebookresearch/fairseq/tree/main/examples/wav2vec/unsupervised/kaldi_self_train`

minimize the NLL while simultaneously maximizing vocabulary usage. The NLL metric reflects the model's proficiency in generating sensible sentences consistent with the dataset distribution, while vocabulary usage quantifies the diversity of phonemes employed by the model. This combined approach ensures that our unsupervised ASR system's predictions align with the expected LM distribution and exhibit a wide linguistic range, thus mitigating the risk of favoring repetitive, high-probability sentences.

Additionally, the lengths of transcriptions are taken into account. We employ NLL without length normalization, thereby favoring predictions that score high under the language model but without an undue preference for overly lengthy transcriptions. The optimization objective is formally expressed as:

$$\theta^* = \arg\min_\theta \left( \frac{-\sum_{i=1}^M \log p_{\text{LM}}(y'_{i,\theta})}{U(Y'_\theta)} \right) \tag{10}$$

## C.6  Training Details

For the training of our segmentation model, we initially employ behavioral cloning, followed by updates using policy gradient methods. In the BC phase, we configure our model with a batch size of 128 and a learning rate of 0.0005, training for 20 epochs. We also assign cross-entropy weights of [1, 5] for boundary classification, to address the imbalance where the class labeled 0 significantly outnumbers the class labeled 1.

During the policy gradient update phase, the model is trained with the same batch size of 128, but with a reduced learning rate of 0.0001 and a weight decay of 0.0001. We utilize the AdamW optimizer in conjunction with the CosineAnnealingLR scheduler. For the LS and MLS datasets, the training proceeds for 40 epochs, while for the TIMIT dataset, owing to its smaller size, we extend the training to 500 epochs to ensure convergence.

To refine the performance of our reinforcement learning framework, we have tuned the reward coefficients, leveraging the validation metric (Eq. 10), to secure optimal outcomes across varied datasets. Specifically, the coefficient for the PPL difference reward, $c_{\text{ppl}}$, is fixed at 1.0 to prioritize the minimization of perplexity in generated sequences. The coefficient for the edit-distance reward, $c_{\text{edit}}$, is set at 0.2. And the coefficient for the length difference reward, $c_{\text{len}}$, is selected from a predefined set of values within the range $[0.0, 0.2, 0.4, 0.6, 0.8]$. These values are carefully chosen to balance the importance of length consistency with other factors, with the optimal reward configuration for each dataset detailed in Table 11.

It is noteworthy that, as the iterations progress, the length constraints become less critical. This observation suggests that the model gradually learns to generate sequences of appropriate length, reducing the need for explicit length-based regularizations.

As for the Stage 2 GAN training, we inherit the three weighted coefficients in Eq. 9 from the previous iteration. The initial training of wav2vec-U takes 150000 steps following the original paper. After the initialization stage, we find out that optimizing with 20000 steps for each iteration is enough for converging, taking advantage of initializing the parameters from the previous stage.

All of the experiments can be done using a single NVIDIA V100 GPU. The initial stage of wav2vec-U GAN training and the REBORN stage 1 reinforcement learning takes about 12 hours of training on

Table 11: Best reward configurations obtained through hyperparameter searches on each dataset.

| Dataset | LibriSpeech | | TIMIT | MLS | | | | | |
|---|---|---|---|---|---|---|---|---|---|
| | iter. 1 | iter. 2-5 | iter. 1-3 | de | nl | fr | es | it | pt |
| $c_{\text{ppl}}$ | 1.0 | 1.0 | 1.0 | 1.0 | 1.0 | 1.0 | 1.0 | 1.0 | 1.0 |
| $c_{\text{edit}}$ | 0.2 | 0.2 | 0.2 | 0.2 | 0.2 | 0.2 | 0.2 | 0.2 | 0.2 |
| $c_{\text{len}}$ | 0.2 | 0.0 | 0.0 | 0.4 | 0.2 | 0.8 | 0.4 | 0.6 | 0.4 |

an NVIDIA V100 GPU for each run. As for REBORN stage 2, since the model is not randomly initialized, it only takes about 1.5 hours of training time.

### C.7 WFST Decoding for Obtaining Word-Level Outputs

In REBORN, phoneme-level outputs can be generated directly with greedy decoding. However, obtaining word-level outputs requires additional modules such as WFST or external language models [4, 18, 34]. In this work, we incorporate REBORN with WFST to generate word-level transcriptions. Our decoding procedure mainly follows Baevski et al. [4], which builds the WFST using PyKaldi [9], and additional self-loops are introduced to mimic the CTC behavior. Moreover, two hyperparameters: the acoustic scale $a$, and the additive blank weight $v$ are added during the WFST decoding process. In wav2vec-U, the search interval of $a$ is $[0, 8]$ and $v$ is tuned within $[-3, 8]$. As for REBORN, we find that using a much smaller search space for both of the hyperparameters is enough for obtaining reasonable outputs. More specifically, We tune the acoustic scale in $[0.4, 1.2]$ with step $0.1$ and the blank weight in $[-5, 2]$ with step $1$.

It is worth mentioning that we utilize the *original boundary* generated by the segmentation model instead of the *merged* one (§ 3.1.5) for WFST decoding. We find that the increased amount of segments directly leads to longer generator outputs, which is beneficial for WFST to generate higher-quality outputs. The phenomenon is also discovered by the prior works [4, 30]. In wav2vec-U [4], the segmental feature used for WFST decoding is the one before adjacent pooling; while in wav2vec-U 2.0 [30], they simply lower the decoding stride to generate more lengthy outputs for WFST decoding.

## D  Speech Self-supervised Learning and Acoustic Unit Discovery

Self-supervised learning (SSL) aims to explore effective ways for representation learning without labeled data. Recent speech SSL models can be categorized into the three classes based on the pretraining objective [37], including generative [29, 31, 32], contrastive [2, 3, 43], and predictive [5, 13, 23]. The SSL representation can be utilized for many different downstream tasks, such as phoneme recognition, speaker identification, or emotion recognition [62]. Recent UASR also takes advantage from speech representations from SSL models [4, 18].

Extended from self-supervised learning, acoustic unit discovery mainly focuses on deriving phoneme or word-like units or learning speech representations that retain only linguistically relevant information in an unsupervised manner [16, 17, 28, 40, 42]. The task is highly related to unsupervised phoneme/word segmentation, and the standard evaluation protocols include the phoneme/word boundary F1, frame-level ABX score [53] on ZeroSpeech Challenge [40], or phoneme accuracy with **supervised learned** linear prediction head [16]. Although both acoustic unit discovery and unsupervised ASR are learned without supervision, they differ a lot in their learning objectives. As an extended topic of representation learning, acoustic unit discovery targets on learning better-segmented representations; while unsupervised ASR directly aims at solving the distribution matching problem, and may utilize high-quality speech representations to reach the goal.

## E  Different Schemes for Boundary Evaluation

According to Strgar and Harwath [56], the original protocol for evaluating phoneme boundaries contains some ambiguity. The issue arises from double counting when both the ground truth boundaries and the predicted boundaries fall within the same tolerance window. To address this issue, they propose a *harsh* (or *strict*) evaluation protocol that prevents double counting. The original protocol is referred to as the *lenient* boundary evaluation protocol. It can be assumed that studies conducted before Strgar and Harwath [56] used the lenient evaluation method, which often overestimated the quality of the predicted boundaries. We encourage interested readers to refer to the original paper for more detailed explanations.

