# OpenReview forum: "REBORN: Reinforcement-Learned Boundary Segmentation with Iterative Training for Unsupervised ASR"
_NeurIPS.cc/2024/Conference — NeurIPS 2024 poster_

### Official Review · Reviewer_q9jx · 2024-07-05

**Soundness:** 3
**Presentation:** 2
**Contribution:** 3
**Rating:** 7
**Confidence:** 2

**Summary:**

This paper proposes a training framework for unsupervised speech recognition models, building on top of wav2vec-u. The authors note that the performance in wav2vec-u is hindered by the quality of segment boundaries, as they show that the phoneme-error-rate can be improved significantly with ground truth segmentation boundaries. The proposed method introduces a segment boundary prediction mode and a phoneme prediction model. First, the segment boundary prediction model is trained with a policy gradient method. The reward function is based on the perplexity of the phoneme sequence. This sequence is predicted from the computed segmentation boundaries, and is compared to earlier versions of the boundary prediction model, i.e, the boundary segmentation model from a previous iteration, or wav2vec-u at iteration 0. The perplexity should be as low as possible and can be computed by using a language model from unpaired text data. Secondly, the phoneme prediction model is trained with an adversarial loss, where the phoneme prediction model acts as a generator of a phoneme sequence, and the discriminator has to distinguish between these and real phoneme sequences from the unpaired text data. These steps as then repeated until no performance improvements are observed.

**Strengths:**

The authors clearly show that the wav2vec-u model suffers from the quality of the segmentation boundaries, and propose a well-described methodology to improve on it. The authors report results on the expected datasets, and perform a small ablation study to analyze the behavior of the proposed method.

**Weaknesses:**

The authors propose a *very* complex methodology, with a lot of moving and finicky components, as reinforcement learning and GANs are both known to be fickle and hard to reproduce. I worry therefore that it will be hard to reproduce these results, but the authors promised in the checklist to release the code.

The authors use many tricks to make the method more stable and performant:
1. Initialize the phoneme prediction model with the pre-trained wav2vec-u model
2. Introduce 2 auxiliary reward functions, which heavily punish predicted sequences which are longer or shorter than the previous iteration
3. Use behavior cloning (supervised loss on boundary pseudolabels) before the RL loss
4. boundary merging (segments with the same phoneme prediction are merged)
5. self-training of the phoneme prediction model (supervised loss on phoneme pseudolabels)

These are not a weakness per se, but point towards the overall complex picture of the proposed method. I hope future work can find a simpler and more elegant formulation of the UASR problem :)

The authors claim their method is iterative, but in paragraph 4.3 state they only use 1 to 2 iterations. I then wonder why the authors try to sell this as an iterative method, instead of follow-up stages to (significantly) improve on wav2vec-u.

Minor comments on writing:
* line 21: supervisedly trained with a ... -> trained with self-supervision using a ...
* line 33: translate and -> translate, and
* line 50: model: -> model.
* line 74: remove "the"
* line 77: GAN -> The GAN
* line 95: The wav2vec-U network
* line 98: to -> for
* line 99: I cannot really parse this sentence, origanated to be -> originally (?)
* line 102: We discuss these two related topics further in Appendix F.
* line 149: The REINFORCE algorithm

**Questions:**

Why do you state in 4.3 to only use 1 to 2 iterations, while figure 2 shows results with improvements until 5 iterations?

Why do you think that your model performs better on 5 languages but worse on Portuguese compared to wav2vec-u 2.0?

Why did you decide to mean-pool the segment before predicting the phoneme? I would venture that this throws away information and is not necessary?

**Limitations:**

The ultimate goal of UASR is to enable speech technology for digitally rare languages without good labeled data. Although this work studies UASR on 6 languages, they are all from rich European countries. It remains to be seen whether this method works for languages from a different continent, especially because this method relies heavily on high quality textual data, to compute the perplexity in the RL loss, and for the discriminator.

---

> ### Author Rebuttal · Authors · 2024-08-06
>
> We sincerely thank the reviewer for the thorough and constructive comments. Please find the response to your questions below.
>
> > The authors propose a very complex methodology. **These are not a weakness per se**, but point towards the overall complex picture of the proposed method.
>
> To ensure the reproducibility of our work, **we will release the codes, training and evaluation scripts, and model weights**.
>
> > Why the authors try to sell this as an iterative method, instead of follow-up stages to (significantly) improve on wav2vec-u.
>
> > Why do you state in 4.3 to only use 1 to 2 iterations, while figure 2 shows results with improvements until 5 iterations?
>
> While the performance gain is significant in the early iterations, Figure 2 shows that further improvements can still be achieved with more iterations. **In Table 1, we use only two iterations to emphasize the effective integration of REBORN and self-training,** as well as to demonstrate the efficiency of REBORN, i.e., achieving improved performance only requires few iterations.
>
> > Minor comments on writing:
>
> We thank the reviewer for the detailed suggestions. We will revise the paper to fix them.
>
> > Why do you think that your model performs better on 5 languages but worse on Portuguese?
>
> We hypothesize that our method performs slightly worse in Portuguese because the cross-lingual feature extractor (XLSR-53) uses the least amount of Portuguese data compared to other languages during self-supervised pre-training.
>
> > Why did you decide to mean-pool the segment before predicting the phoneme?
>
> Our phoneme prediction model is initialized from wav2vec-U, which uses mean-pooled features. To ensure consistency with wav2vec-U, we also use mean-pooling in REBORN. Exploring other pooling methods is interesting and potentially beneficial, which is left for future work.
>
> > The ultimate goal of UASR is to enable speech technology for digitally rare languages without good labeled data. Although this work studies UASR on 6 languages, they are all from rich European countries.
>
> We thank the reviewer for pointing this out. Indeed, we have not yet tested REBORN on a digitally rare language. Referencing prior works [1-3], we tested REBORN on the most commonly used datasets in UASR. We will revise the paper to include this limitation in Appendix A.
>
> ### References
>
> [1] Chen et al. “Completely Unsupervised Phoneme Recognition by a Generative Adversarial Network Harmonized with Iteratively Refined Hidden Markov Models.” *Interspeech, 2019*.
>
> [2] Baevski et al. "Unsupervised speech recognition." *NeurIPS, 2021*.
>
> [3] Gao et al. "Euro: Espnet unsupervised asr open-source toolkit." *ICASSP, 2023*.

---

> > ### Comment · Reviewer_q9jx · 2024-08-12
> >
> > I acknowledge the rebuttal and will not be changing my (favorable) score.

---

### Official Review · Reviewer_AM1A · 2024-07-12

**Soundness:** 4
**Presentation:** 4
**Contribution:** 4
**Rating:** 7
**Confidence:** 4

**Summary:**

This paper addresses the challenging problem of learning speech recognition without parallel recordings and transcriptions. They build upon state-of-the-art approaches of Wav2Vec-U(2) in this area and propose to refine a pretrained model using a reinforcement learning approach. The main idea is to split the problem into two parts: segmentation and segment classification into “phonemes”. Since neither segmentation nor labels are known, they use policy gradients and leverage “phoneme” classifier rewards to learn the segmentation model. The training is stabilized by splitting it into iterations where only one of the two models is being trained while the other is frozen, and additional rewards (penalties?) that act as regularizers for the main perplexity difference reward. In essence, the model is overall encouraged to improve the perplexity vs the previous model checkpoint, but not change either the segment length or phone edit distance too much. The evaluation is performed on 100h subset of LibriSpeech and full TIMIT for English, and 100h subsets of MLS for German, Dutch, French, Spanish, Italian and Portuguese. The authors demonstrate significant improvements in WER and PER vs state-of-the-art baseline models.

**Strengths:**

* Their main idea of framing UASR as an RL problem is original and interesting. Overall I think it is a solid contribution to UASR literature.
* The proposed RL training is able to significantly improve upon Wav2Vec-U ASR results.
* The evaluation setup is comprehensive enough to believe the method may be generally applicable for other languages.
* The method, architecture, and experimental setup are described comprehensively.
* The ablations related to reward function designs are interesting and demonstrate that the proposed edit-distance and length-difference rewards are effective regularizers for the main perplexity difference reward.

**Weaknesses:**

* The author’s (mis)use of the word “phoneme” is problematic. E.g., p.2 l.64 mentions “segmental structures (that) are acoustic units smaller than phonemes”. Phoneme is a perceptual construct that represents all possible (sequences of) sounds (phones) for which there is no lexical distinction. Such sequence may very well be empty. Therefore it is problematic to talk about phonemic segmentation; what the authors may have had in mind is phonetic segmentation instead. This problem is discussed more thoroughly in: Moore, R. K., and L. Skidmore. "On the use/misuse of the term 'phoneme'." Proceedings of Interspeech 2019.
* The authors claim that “the effect of feature segmentation on phoneme prediction is non-differentiable”, however, it can be. The authors may consult e.g. the segmental CPC approach with its differentiable boundary detection in: Bhati, S., Villalba, J., Żelasko, P., Moro-Velázquez, L., Dehak, N. (2021) Segmental Contrastive Predictive Coding for Unsupervised Word Segmentation. Proc. Interspeech 2021, 366-370, doi: 10.21437/Interspeech.2021-1874

**Questions:**

* Could REBORN be leveraged in a semi-supervised setup, e.g., starting from an ASR pretrained on a low amount of supervised data, to leverage a larger amount of non-parallel data for fine-tuning?

**Limitations:**

* The proposed method depends strongly on the availability of an initial pretrained ASR model (presumably unsupervised). My understanding is that it is more of a “refinement” stage for existing models rather than stand-alone method. I think it should be presented as such - the title and the abstract don’t mention that.

---

> ### Author Rebuttal · Authors · 2024-08-06
>
> We sincerely thank the reviewer for the thorough and constructive comments. Please find the response to your questions below.
>
> > The author’s (mis)use of the word “phoneme” is problematic. What the authors may have had in mind is phonetic segmentation instead.
>
> We thank the reviewer for pointing this out. **We will thoroughly revise the paper to ensure the correct use of these terms**.
>
> > The authors claim that “the effect of feature segmentation on phoneme prediction is non-differentiable”, however, it can be.
>
> We would like to clarify that in our paper, we meant that the process of predicting phonemes based on predicted segments is naturally non-differentiable. As the reviewer pointed out, some techniques could allow for approximating the backpropagation gradients. **We will revise our claim in the paper to avoid confusion and discuss the work [Bhati et al., 2021] referred by the reviewer**.
>
> > Could REBORN be leveraged in a semi-supervised setup?
>
> We thank the reviewer for this interesting idea. We believe that REBORN could benefit especially when the labels of the limited supervised data are noisy. In this way, the phoneme predictor might be non-ideal and could possibly get improved through REBORN’s iterative paradigm. We would love to investigate this idea in the future thoroughly.
>
> > It should be presented as a “refinement” stage for existsing models, but the title and the abstract don’t mention that.
>
> We will rethink the paper's title and abstract to clarify this. We appreciate the reviewer’s suggestion.

---

> > ### Comment · Reviewer_AM1A · 2024-08-12
> >
> > Thank you for your response. I uphold my recommendation to accept.

---

### Official Review · Reviewer_vhRd · 2024-07-12

**Soundness:** 3
**Presentation:** 3
**Contribution:** 3
**Rating:** 7
**Confidence:** 4

**Summary:**

The paper proposes an improvement over wav2vec-U for unsupervised speech recognition, specifically for the task of predicting phonemes. For the final WER performance, a given lexicon is used to get from the phoneme sequence to words.

So the paper focuses on improving the segmentation/boundaries of the phonemes.

Two model parts:

The segmenter, trained via RL and various different reward functions.

GAN-style training for the phoneme prediction model based on the segments from the segmenter.

Stage 1 trains only the segmenter, and assumes a given frozen phoneme prediction model. In the first iteration, the phoneme prediction model is initialized from  wav2vec-U.

Stage 2 trains only the phoneme prediction model, taking the segments from the segmenter.

In GAN-style training of the phoneme prediction model: Phoneme prediction model is the generator, and discriminator tries to distinguish a generated/predicted phoneme sequence vs a real one, using the unpaired text data.

**Strengths:**

Good results.

Code will be published.

**Weaknesses:**

Some parts could be made more clear. (See below.)

Needs Wav2vec-U for initialization.

**Questions:**

So you train a phoneme generator model here. But it's a bit unclear, how do you get from phonemes to words? It refers to the appendix, and WFST decoding is mentioned. It mentions that it needs a lexicon. So to make this clear: the lexicon provides a given mapping from phonemes to words? If this is given, is it fair to call this unsupervised ASR then? It seems like you cheat here by giving it a crucial part for the unsupervised ASR task. While this is still an interesting problem to study then, I think this should be made much more clear. This is not really unsupervised ASR to me. Or if you really want to stick to the term, at least make it very clear (not just in the appendix, and only for readers who know what a "lexicon" is) that this is what you take as a given here. This also has further implications: The phoneme inventory is then also given.

The exact definition of the phoneme predictor/generator is a bit unclear. Is this an auto-regressive model? Or non-autoregressive?

How do you sample from the phoneme predictor for policy gradient? Does this involve beam search?

**Limitations:**

-

---

> ### Author Rebuttal · Authors · 2024-08-06
>
> We sincerely thank the reviewer for the thorough and constructive comments. Please find the response to your questions below.
>
> > How do you get from phonemes to words? ... it needs a lexicon.  If lexicon is used, is it fair to call this unsupervised ASR? I think this should be made much more clear.
>
> We thank the reviewer for raising this concern. **Our work does utilize an additional phonemizer (L668-L670) to obtain lexicon, following the standard setup in UASR [1-5]**. We acknowledge that using a lexicon is a limitation of recent UASR works, including ours, which is discussed in Appendix A (L540).  As suggested by the reviewer, **we will revise the paper to clearly discuss this in Section 2 and Section 4.3**.
>
> > The definition of the phoneme predictor is unclear. Is it autoregressive or non-autoregressive?
>
> It is non-autoregressive. Precisely, it is a one-layer CNN following wav2vec-U. We will revise the paper to include this information in Section 4.3 to make it clear.
>
> > How do you sample from the phoneme predictor for policy gradient? Does this involve beam search?
>
> When training the segmentation model using policy gradient, we obtain the phoneme predictor by taking the argmax prediction from the phoneme predictor.  Since our phoneme prediction model is non-autoregressive, we do not perform a beam search. We will revise the paper to include this detail in Section 3.
>
> > Needs wav2vec-U for initialization
>
> We acknowledge that relying on a reasonably effective phoneme prediction model for initialization may present a limitation, as noted in L539. However, we would like to clarify that our method is not limited to the wav2vec-U for initialization; instead, we have experimented with using different feature extractors, such as HuBERT [6] and WavLM [7] following EURO [4], and starting from different phoneme predictors on LibriSpeech (see Appendix C.5). The results show that **REBORN achieves comparable performance improvements even with a completely different backbone model and a less effective phoneme predictor**.
>
>
> ### References
>
> [1] Liu et al. “Completely Unsupervised Phoneme Recognition by Adversarially Learning Mapping Relationships from Audio Embeddings.” *Interspeech, 2018*.
>
> [2] Yeh et al. "Unsupervised speech recognition via segmental empirical output distribution matching.” *ICLR, 2019*.
>
> [3] Baevski et al. "Unsupervised speech recognition." *NeurIPS, 2021*.
>
> [4] Gao et al. "EURO: Espnet unsupervised asr open-source toolkit." *ICASSP, 2023*.
>
> [5] Liu et al. “Towards end-to-end unsupervised speech recognition.” *SLT, 2022*.
>
> [6] Hsu et al. "Hubert: Self-supervised speech representation learning by masked prediction of hidden units." *TASLP, 2021*.
>
> [7] Chen et al. "Wavlm: Large-scale self-supervised pre-training for full stack speech processing." *JSTSP, 2022*.

---

> > ### Comment · Reviewer_vhRd · 2024-08-12
> >
> > Thank you for the answer. With those points being made more clear in the paper, I will increase my rating by one to 7 (Accept).

---

### Official Review · Reviewer_TC8W · 2024-07-14

**Soundness:** 3
**Presentation:** 4
**Contribution:** 3
**Rating:** 6
**Confidence:** 3

**Summary:**

This paper proposes a new UASR approach which explicitly learns both segmentation (phoneme boundary) prediction and phoneme class prediction. To do segmentation, the paper proposes some reinforcement-learning objectives. The phoneme class prediction follows an existing approach. It turns out the proposed approach is effective, achieving the new SOTA for UASR on public datasets.

**Strengths:**

- The paper's presentation is clear and is easy to follow.
- The proposed technique (RL for learning segmentation model) is novel and non-trivial
- The proposed technique is effective. It achieves SOTA by a big margin across several datasets and languages.
  - There is analysis why the proposed technique works so effectively
  - See my concern later

**Weaknesses:**

- In fact, ASR don't need accurate segmentation, e.g. CTC model is peaky -- the peak of a phoneme can appear at any frame for this phoneme. For this reason, it is not so clear to me why improving segmentation/phoneme granularity can help improve UASR so much.
  - Please justify.
  - For this reason, I put the rating to "border line" instead of "weak accept"

- Unfortunately, the predicted segments are not more accurate than other approaches and probably are not good to be used to obtain accurate timestamps for ASR (or not? -- you may use TIMIT or Buckeye's ground-truth timestamp to check this)

- L324:  "some segmental structures smaller than the phonemes": does it probably mean that -- if we use sub-phoneme structures in wav2vec-U 2.0, it may be as performant as the proposed approach, as we are able to learn a smaller granularity of signals. In this case, we won't need the RL learning.
  - Would be better to give a real example

**Questions:**

- Besides Table 4, it would be better to add a table for TIMIT which comes with word/phone-level timestamps. Another dataset Buckeye can be considered.
- Typo or presentation suggestions:
  - Figure 1: probably add "1 means the start of a segment" to the caption to make it self-contained

**Limitations:**

Yes.

---

> ### Author Rebuttal · Authors · 2024-08-06
>
> We sincerely thank the reviewer for the thorough and constructive comments. Please find the response to your questions below.
>
>
> > ASR don't need accurate segmentation, e.g. CTC model is peaky -- the peak of a phoneme can appear at any frame for this phoneme.
>
> While recent **supervised ASR** does not need explicit segmentation, under the **unsupervised scenario**, learning the mapping from very long speech features to short text transcriptions has been found to be very challenging. To mitigate this difficulty, **segmentation is critical to ensure good unsupervised ASR performance and stability**, which is evident from early works [1-3] to recent studies [4-5].
>
> To investigate the importance of segmentation in UASR, we compare the training stability of UASR algorithms: wav2vec-U, wav2vec-U 2.0, and REBORN. wav2vec-U 2.0 is the only UASR algorithm that does not segment the speech features. wav2vec-U uses k-means segmentation (see Appendix D), while REBORN uses a segmentation model learned using RL. Following the EURO setup described in Section 4.1, we use each algorithm to train UASR models on LibriSpeech. Since each algorithm has its own configuration and search space, including the random seeds, we iterate through these configurations, with each resulting in one model. For each algorithm, we report the *"percentage of models yielding PER < 40%"* among all training configurations and random seeds. The results are shown in Table 1 below.
>
> *Table 1: The importance of segmentation in current UASR. Results are obtained from LibriSpeech following the EURO setup. As the table indicates, **wav2vec-U and REBORN are fully converged in PER with proper segmentation**.*
>
> | Method | &nbsp; Segmentation | &nbsp; Number of models we trained &nbsp;&nbsp;&nbsp; | Percentage of models with PER < 40% ↑ |
> |-|:-:|:-:|:-:|
> | wav2vec-U (k-means-based) | ✓ | 40 | 100% |
> | REBORN | ✓ | 50 | 100% |
> | wav2vec-U (no-segmentation) | x | 40 | 0% |
> | wav2vec-U 2.0 | x | 64 | 19% |
>
> From Table 1, we observe that wav2vec-U 2.0, which does not have segmentation, is highly unstable. Conversely, REBORN and wav2vec-U (with k-means segmentation) always result in a UASR model with PER lower than 40%. Additionally, removing the segmentation from wav2vec-U makes the algorithm unstable and never yields a model with PER lower than 40%.
>
> > For this reason, it is not so clear to me why improving segmentation/phoneme granularity can help improve UASR so much.
>
> The results in Table 1 have justified the importance of segmentation in UASR. Moreover, we found that the quality of the segmentation strongly affects the performance of UASR (Table 4 in the paper): **using hand-crafted rules or separately learned segmentation boundaries yields suboptimal UASR performance while using the oracle phoneme boundary greatly improves the UASR performance**. Motivated by the above observation, we thus propose to use RL to learn segmentation that is **tailored for the phoneme prediction model**. The segmentation boundary in REBORN is learned with feedback from the phoneme prediction model, which ensures the boundary is useful for improving the UASR performance and **attains state-of-the-art results on many widely used UASR datasets**.
>
> > The predicted segments are not more accurate than other approaches and probably are not good to be used to obtain accurate timestamps.
>
> As suggested by the reviewer, we use TIMIT’s human-annotated phone-level timestamps for evaluation and report the results in Table 2 below.
>
> *Table 2: Boundary evaluation results on TIMIT. REBORN with boundary merging (Figure 1-(c\)) is close to the existing SoTA method.*
>
> |Method|Precision|Recall|F1 Score|R-Value|
> |-|-|-|-|-|
> |k-means-based|0.62|0.75|0.68|0.68|
> |Strgar and Harwath [6]|0.85|0.79|0.82|0.84|
> |REBORN|0.61|0.83|0.71|0.62|
> |REBORN (w/ boundary merging)|0.80|0.78|0.79|0.82|
>
> The results show that the initial boundaries learned by REBORN (before boundary merging in Figure 1-(c)), which are specifically optimized for the phoneme prediction model, already achieve a high recall. With boundary merging, i.e., consecutive segments with the same phoneme prediction are merged as illustrated in Figure 1-(c\), **REBORN achieves better alignment, bringing our results close to existing SoTA unsupervised segmentation methods**.
>
> > It would be better to add a table for TIMIT which comes with word/phone-level timestamps.
>
> We thank the reviewer for suggesting this experiment, which strengthens the contribution of our method. We will revise the paper to include this result.
>
> > if we use sub-phoneme structures in wav2vec-U 2.0, it may be as performant as the proposed approach
>
> Since wav2vec-U 2.0 uses a fixed stride 1D-CNN to downsample the speech features and does not segment the speech features, we are unsure how to incorporate sub-phoneme structures in wav2vec-U 2.0. We would appreciate it if the reviewer could provide more details. We are willing to implement, analyze, and discuss it if time allows.
>
> > Figure 1 presentation suggestions
>
> We thank the reviewer for the suggestion. We will revise Figure 1 based on the suggestion.
>
> ### References
>
> [1] Liu et al. “Completely Unsupervised Phoneme Recognition by Adversarially Learning Mapping Relationships from Audio Embeddings.” *Interspeech, 2018*.
>
> [2] Yeh et al. "Unsupervised speech recognition via segmental empirical output distribution matching.” *ICLR, 2019*.
>
> [3] Chen et al. “Completely Unsupervised Phoneme Recognition by a Generative Adversarial Network Harmonized with Iteratively Refined Hidden Markov Models.” *Interspeech, 2019*.
>
> [4] Baevski et al. "Unsupervised speech recognition." *NeurIPS, 2021*.
>
> [5] Gao et al. "Euro: Espnet unsupervised asr open-source toolkit." *ICASSP, 2023*.
>
> [6] Strgar et al. “Phoneme segmentation using self-supervised speech models.” *SLT, 2022*.

---

> > ### Comment · Reviewer_TC8W · 2024-08-13
> > **Acknowledgement of rebuttal**
> >
> > Thank you for the explanation and details in the rebuttal. With the explanation of that segmentation is critical for UASR, I'll increase my rating.

---

### Comment · Area_Chair_kewG · 2024-08-12
**Rebuttal and discussion needed**

Hi folks,

AC here. I'm trying to encourage all parties (authors and reviewers) to chip in, since there are some disagreement among all.

I have read the paper and all the reviews. Here are three points I would like to hear from you.

1. It's unclear how the approach is different from Cuervo et al. (2022) (cited as [15] in the submission). Given that the authors are aware of this paper, the paper clearly fails to address the novelty (also raised by reviewer AM1A). Even the reward functions are very similar to the ones in Cuervo et al. (2022).

Cuervo et al., Variable-rate hierarchical CPC leads to acoustic unit discovery in speech, 2022

2. The positive reviews are mostly due to the strong results, while being sparse on the soundness of the paper and where the improvement is coming from. Reviewer TC8W is not convinced that the improvement is coming from segmentation, while reviewer q9jx and AM1A think the improvement is from the iterative training.

3. It's a little surprising to see so few segmentation results except Table 4 (also raised by reviewer TC8W), given how much the emphasis is on segmentation.

The results in Table 4 are weird. 1) Worse segmentation results actually lead to better PERs, contradicting to the claim in the paper that better segmentation leads to better PERs. 2) The proposed approach is over-segmenting a lot and does not seem to perform well compared to something as simple as k-means. 3) The paper deliberately emphasizes that the result is without boundary merging. Why not report both with and without? 4) How come using the oracle segmentation leads to a worse PER?

The segmentation results on TIMIT (in the reply to reviewer TC8W) are useful, but don't seem to answer the questions around Table 4.

Thanks.

---

> ### Author Response · Authors · 2024-08-13
> **Re: Rebuttal and discussion needed (1/2)**
>
> We sincerely thank the AC for the thorough comments. Please find the response to your questions below.
>
> > It's unclear how the approach is different from Cuervo et al. (2022) (cited as [15] in the submission). Given that the authors are aware of this paper, the paper clearly fails to address the novelty (also raised by reviewer AM1A). Even the reward functions are very similar to the ones in Cuervo et al. (2022).
>
> Our work significantly differs from Cuervo et al. (2022) [15] in the following aspects:
> - **Task**: The method presented in [15] is designed for **acoustic unit discovery**, whose goal is to learning better-segmented representations. On the other hand, our work aims to address **unsupervised ASR**, which aims to directly solve the distribution matching problem between speech segments and phoneme sequences. The differences between these two tasks are specifically discussed in Section F of the Appendix.
> - **RL reward**: The reward proposed in [15] uses contrastive loss (Equation 2), $\mathcal{L}\_H = - \sum\_{k} \sum\_{m=1}\^{M} \frac{\exp(p\_m\^T u\_{k+m}\^{q})}{\sum\_{i \in \set{k+m-1, k+m+1}} \exp(p_m^T u_i^q)} $, which drastically differs from our reward design. Specifically, we design the perplexity difference reward calculated by a phoneme language model, along with the edit distance reward and the length difference reward as regularization rewards: $R = c_{\text{ppl}} \cdot R_{\text{ppl}}  + c_{\text{edit}} \cdot R_{\text{edit}} + c_{\text{len}} \cdot R_{\text{len}}$, where $R_{\text{ppl}} = \text{PPL}\_{\theta-1} - \text{PPL}\_{\text{$\theta$}}$, $R_{\text{edit}} = -\frac{d\_\text{Lev}(Y'\_{\theta-1}, Y'\_{\theta})}{|Y'\_{\theta-1}|}$, and $R_{\text{len}} = 1 - \frac{\left| |Y'\_{\theta}|  - |Y'\_{\theta-1}| \right|}{|Y'\_{\theta-1}|}$.
>
> Our work and [15] only share a similar high-level idea, which is using RL for boundary prediction. The main contribution of our work compared to [15] is that we find that in UASR, the segmentations that are tailored for the phoneme prediction model are crucial to the PER, and we use RL to allow the segmentation model to tailor the segmental boundary for the phoneme prediction model. Designing rewards for UASR to learn a segmentation model is non-trivial, and the reward design is completely different from [15] and not seen in any prior works in acoustic unit discovery.
>
> > The positive reviews are mostly due to the strong results, while being sparse on the soundness of the paper and where the improvement is coming from.
>
> > Reviewer TC8W is not convinced that the improvement is coming from segmentation, while reviewer q9jx and AM1A think the improvement is from the iterative training.
>
> We would like to point out that the numeric scores of soundness in the review form show that all four reviewers consider this paper to be sound, with soundness scores 3, 3, 3, and 4. Moreover, in the latest response to the author’s rebuttal about segmentation, Reviewer TC8W agrees with us on the criticality of segmentation in UASR.
>
> We believe that REBORN improves PER, compared to wav2vec-U, mainly from the segmentation model. Specifically, REBORN trains the segmentation model to tailor the segmental boundary. Under an unsupervised ASR setting, learning the segmental boundary for the phoneme prediction model without ground truth transcription is highly non-trivial. To overcome this difficulty, we construct carefully designed rewards to guide the segmentation model. We say the segmental boundary is *tailored for* the phoneme prediction model because the segmentation model is trained by the guidance of the phoneme prediction model (and with the external LM).  With extensive experimental results presented in Table 4 and the rebuttal’s Table 1, we have verified that proper segmentation tailored to the phoneme prediction model is critical to the UASR performance, which is acknowledged by Reviewer TC8W. Also, Reviewer AM1A considers our reward design to be interesting and effective, which is listed as a strength of this paper.
>
> Last, we would like to respectfully point out that the reviews from Reviewers AM1A and q9jx do not attribute the improvement of REBORN to iterative training. Reviewer AM1A did not mention the word “iterative”. Reviewer q9jx considers the improvement of REBORN in the early stage to be very significant and questions if it is necessary to frame the method as an iterative training method; this is contrary to attributing the improvement to iterative training.

---

> ### Author Response · Authors · 2024-08-13
> **Re: Rebuttal and discussion needed (2/2)**
>
> > It's a little surprising to see so few segmentation results except Table 4 (also raised by reviewer TC8W), given how much the emphasis is on segmentation.
>
> Our work mainly aims to improve the performance of unsupervised ASR. Hence, the evaluations center around UASR performance. To investigate the intermediate results, i.e., segmentations, Table 4 analyzes the role and performance of segmentation. Specifically, Table 4 reveals some of the interesting findings of the REBORN boundary, such as more frequent segmentation, as discussed in L320-L329. Considering that the readers might also be interested in the boundary evaluation score on the human-annotated datasets, we will revise the paper to include the rebuttal’s Table 2 in response to Reviewer TC8W.
>
> > The results in Table 4 are weird. 1) Worse segmentation results actually lead to better PERs, contradicting to the claim in the paper that better segmentation leads to better PERs.
>
> Although our method is motivated by the idea that “a better segmentation can lead to a better UASR performance,” we consider that there are two factors that affect the “*goodness*” of segmentation in UASR:
> - The quality of the segmentation, which can be evaluated by boundary metrics.
> - The suitability of the segmentation for the unsupervised phoneme predictor.
>
> While the oracle boundary in Table 4 has better boundary F1 than the REBORN boundary, it is not tailored for the unsupervised phoneme prediction model. Note that the phoneme prediction model used in Table 4 is obtained from wav2vec-U, which uses the k-means boundary when training the phoneme prediction model. As a result, naively replacing the k-means boundary with the oracle boundary cannot yield the optimal PER. Contrarily, the segmental boundary learned by REBORN is better suited for the phoneme prediction model, yielding a better PER.
>
> > 2\) The proposed approach is over-segmenting a lot and does not seem to perform well compared to something as simple as k-means.
>
> The lower F1 score for REBORN compared with k-means is because the segmentation is tighter. However, over-segmentation is not an issue since most segments result in consecutive phoneme predictions, which do not affect PER as they are de-duplicated, as noted in the paper (L132-L133). Phoneme deduplication is a common practice in UASR for generating more reasonable results ([1-3]), and is also involved in the training process of REBORN.
>
> > 3\) The paper deliberately emphasizes that the result is without boundary merging. Why not report both with and without?
>
> Adopting boundary merging can lead to higher boundary F1, but in Table 4, we focus on the original RL-learned boundary to emphasize the importance of tailoring segmentation for the unsupervised phoneme predictor. Merging boundaries directly changes the segmentation, making it less suitable for the current phoneme predictor. This technique is introduced to stabilize stage 2 training rather than boost boundary F1 performance (see REBORN stage 2 in Table 7, where PER std is larger without boundary merging). Nevertheless, to provide a comprehensive boundary evaluation, we will include Table 2 from the rebuttal in our next revision.
>
> > 4\) How come using the oracle segmentation leads to a worse PER?
>
> As introduced earlier, given that **the phoneme predictor is non-ideal**, directly using the oracle segmentation, although it still greatly improves the PER, may not be the optimal solution. REBORN segmentation, which is **tailored for the non-ideal phoneme predictor**, results in better improvements. However, referring to the first rows of Table 1 and Table 2, if we train the phoneme predictor directly on the oracle segmentation, the GAN-trained predictor is more suitable with the oracle boundary, thus surpassing the performance of REBORN.
>
> ### References
>
> [1] Baevski et al. "Unsupervised speech recognition." *NeurIPS, 2021*.
>
> [2] Gao et al. "Euro: Espnet unsupervised asr open-source toolkit." *ICASSP, 2023*.
>
> [3] Liu et al. “Towards end-to-end unsupervised speech recognition.” *SLT, 2022*.

---

### Decision · Program_Chairs · 2024-09-25

**Decision:**

Accept (poster)

**Comment:**

I personally am not convinced after prompting the discussion among authors and reviewers, but I'm trusting the reviewers' judgment and recommend acceptance.

I find the numbers reported in the paper sloppy, if not fishy. There is a clear intention to hide the weaker results. It's understandable, but overly practiced in this paper. We, as a community, have lost the opportunity to actually know where the improvement comes from. The paper could have been a lot better if it were open about the weaknesses and where it performs worse. It's not only for better science, but opens up opportunity for others to improve.

I recommend revising the paper to make the experiments controlled, i.e., changing one thing and only one thing at the time (especially between wav2vec-U and iteration 1). In addition, reviewers provide valuable suggestions that lead to proposed changes in the rebuttal. All of these will make this a great paper, and I sincerely hope that the authors make their best effort to include these suggestions.